# Geometric Reciprocity: Unlocking Self-Supervision for Stereoscopic Video Generation

**Jingyi Lu**[1]     **Kai Han**[1]

## Abstract

Monocular-to-stereo conversion synthesizes stereoscopic content from 2D videos for immersive 3D experiences. In modern Depth-Image-Based Rendering (DIBR) approaches, stereo inpainting of disocclusions is the critical bottleneck. Training-based methods achieve superior quality but rely on scarce stereo pairs or synthetic data with domain gaps. We address this through the first self-supervised framework learning from monocular videos via cycle consistency. Our key contribution is the **Geometric Reciprocity Theorem (GRT)**: under the nearest-neighbor DIBR formulation, the disocclusion mask when synthesizing a target view equals the mask of pixels lost when warping back from target to source, enabling analytical computation of test-time disocclusion masks directly from monocular images. This yields train-test consistency for the stated warping formulation, supporting self-supervised learning from unlimited monocular videos and substantial improvements over training-free and supervised state-of-the-art methods. Project page: https://visual-ai.github.io/grt/

## 1. Introduction

The demand for immersive 3D experiences on VR/AR devices, 3D cinema, and stereoscopic displays has made stereoscopic video generation a fundamental problem. The task is to convert monocular videos to stereoscopic format by synthesizing one view from the other (conventionally right-eye from left-eye, or vice versa).

Modern approaches predominantly adopt the Depth-Image-Based Rendering (DIBR) framework (Wang et al., 2024; Shi et al., 2024; Dai et al., 2024; Zhao et al., 2024; Huang et al.,

[1]Visual AI Lab, The University of Hong Kong, Hong Kong, China. Correspondence to: Kai Han <kaihanx@hku.hk>.

*Proceedings of the 43$^{rd}$ International Conference on Machine Learning*, Seoul, South Korea. PMLR 306, 2026. Copyright 2026 by the author(s).

2025; Shvetsova et al., 2026), which decomposes the problem into three sequential stages: estimating depth from the input frame, warping the image to synthesize an initial target view, and inpainting the resulting disocclusions. These disocclusions are regions newly visible in the target view that were occluded in the source, creating geometrically structured missing patterns that general-purpose inpainting methods cannot handle (see Section A7). This domain gap has made the stereo inpainting stage the critical bottleneck.

Due to the scarcity of training data, recent training-free methods (Wang et al., 2024; Dai et al., 2024) manipulate pretrained diffusion models for zero-shot stereo inpainting, but lack stereo-specific geometric priors and produce inferior results. Training-based approaches (Zhao et al., 2024; Huang et al., 2025; Shvetsova et al., 2026) learn these priors yet face a fundamental challenge in training data quality and availability. Existing methods either rely on scarce and often proprietary real stereo pairs with error-prone stereo matching for disocclusion identification (Zhao et al., 2024; Shvetsova et al., 2026), or resort to synthetic data that suffers from inevitable domain gaps (Huang et al., 2025) (see Section A8 for more details).

To address the data bottleneck, we propose the first self-supervised framework that learns stereo inpainting from monocular videos alone, eliminating the need for stereo pairs or synthetic data. Our approach exploits the inherent bidirectional symmetry of stereo view relationships, enabling **right-left-right cycle consistency** where generating the left view from right through DIBR and then reconstructing the right from the synthesized left should recover the original input.

While cycle consistency provides self-supervision in principle, naive implementation requires multiple sequential model inferences and backpropagation through non-differentiable warping operations, making video-scale training computationally prohibitive. We resolve this through a geometric insight: *the stereo warping process itself encodes all information needed to compute cycle consistency losses without executing the cycle.* Through rigorous geometric analysis, we discover that in the right-left-right cycle, the intermediate steps of inpainting the left view, estimating its depth, and performing pixel warping operations are

Table 1. **Comparison of training data construction approaches.** Our Geometric Reciprocity Theorem enables constructing scalable, high-quality training data with train-inference consistent masks from arbitrary real-world monocular videos.

| Preferable Properties | Data Scalability | Real-World Data | Mask Quality | Train-Inference Consistency | Training Efficiency |
|---|---|---|---|---|---|
| Real Stereo Pairs | ✗ | ✓ | ✗ | ✗ | ✓ |
| Synthetic Data | ✓ | ✗ | ✓ | ✓ | ✓ |
| Cycle Consistency (Ours) | ✓ | ✓ | – | ✓ | ✗ |
| **Geometric Reciprocity (Ours)** | ✓ | ✓ | ✓ | ✓ | ✓ |

redundant under the nearest-neighbor DIBR formulation. We formalize this as the **Geometric Reciprocity Theorem (GRT)**: for any right (target) view to be synthesized from a left (source) view, the disocclusion mask required equals the mask of pixels that would be lost when warping back from target to source. This theorem reveals that the cycle consistency supervision signal can be computed analytically from the target view alone using only its depth estimate, with no synthesis, no intermediate views, and no cycling required.

GRT fundamentally transforms training data construction. By treating any monocular image as a target view, we directly compute its *inference-time* disocclusion mask via GRT, yielding the mask induced by the same DIBR geometry used at test time. The image itself then serves as ground truth for these disocclusions, enabling self-supervised training from unlimited monocular videos with train-inference consistency that substantially outperforms both training-free methods and supervised state-of-the-art. We summarize the above comparisons in Table 1.

Our contributions are: (1) the first self-supervised stereo inpainting framework learning from monocular videos without requiring stereo pairs or synthetic data, (2) the Geometric Reciprocity Theorem enabling analytical cycle consistency computation without executing the cycle, (3) the first comprehensive datasets (ImageNet-GRT, Kinetics-GRT, and DAVIS-GRT) with geometrically consistent disocclusion masks for training and evaluation, and (4) state-of-the-art performance surpassing both training-free and supervised methods. Code, precomputed masks, and trained weights will be released at https://github.com/Visual-AI/GRT.

## 2. Related work

### 2.1. Monocular-to-Stereo Video Conversion

Monocular-to-stereo video conversion synthesizes a right-eye view from standard 2D video (assumed as left-eye input) for immersive 3D experiences on stereoscopic displays, 3D cinemas, and VR/AR devices. Early work (Xie et al., 2016) attempted direct regression using end-to-end CNNs but was limited by lack of robust depth priors. Modern

approaches predominantly adopt **Depth-Image-Based Rendering (DIBR)**, which first estimates per-frame depth maps with pretrained models like Depth Anything (Yang et al., 2024), then warps the left view to synthesize an initial right view. This warping uncovers disocclusions—areas occluded in the left view—framing monocular-to-stereo conversion as **stereo inpainting**: filling newly visible regions.

Recent solutions tackle this challenge in various ways. **Training-free methods** leverage pretrained models without fine-tuning. StereoDiffusion (Wang et al., 2024) performs zero-shot inpainting by manipulating latents in pretrained image diffusion models. StereoCrafter-Zero (Shi et al., 2024) uses noisy restart and iterative refinement on video diffusion models. SVG (Dai et al., 2024) contributes a Frame Matrix architecture for video consistency while remaining training-free.

**Training-based methods** demonstrate superior quality and consistency by learning stereo-specific priors from datasets. StereoCrafter (Zhao et al., 2024) fine-tunes Stable Video Diffusion using auto-regressive strategies for temporal coherence. Restereo (Huang et al., 2025) jointly addresses stereo generation and video restoration via training on synthetically degraded data. M2SVid (Shvetsova et al., 2026) improves efficiency through single-step feed-forward prediction conditioned on both original left and warped right views. However, without ready-to-use stereo inpainting datasets, these methods construct training data through preprocessing pipelines. Methods using real stereo pairs face stereo matching errors and copyright restrictions, while those using synthetic data face domain gaps between rendered and real videos, limiting generalization.

### 2.2. Monocular Depth Estimation

Monocular Depth Estimation (MDE) infers dense depth maps from single images. Early methods trained on single-domain datasets like KITTI (Geiger et al., 2012) or NYU-D (Silberman et al., 2012) performed well in specific scenarios but lacked generalization. Recent foundation models achieve zero-shot capabilities through multi-dataset aggregation (MiDaS (Ranftl et al., 2020)) and massive unlabeled datasets (Depth Anything (Yang et al., 2024), Depth-Pro (Bochkovskii et al., 2025)). Advances include genera-

tive models like Marigold (Ke et al., 2024), Vision Transformer architectures (Ranftl et al., 2021), and robustness improvements under adverse conditions (Kong et al., 2023; Zheng et al., 2023; Sun et al., 2025). For DIBR-based monocular-to-stereo conversion, predicted depth maps govern warping by defining per-pixel horizontal displacement, with closer objects shifted more than distant ones to create stereoscopic effects.

### 2.3. Cycle Consistency

Cycle consistency enables unsupervised learning by enforcing bidirectional consistency ($A \rightarrow B \rightarrow A$). CycleGAN (Zhu et al., 2017) demonstrated this for unpaired image-to-image translation. Applications span domain adaptation (Hoffman et al., 2018; Singh, 2021), temporal learning (Dwibedi et al., 2019; Yang et al., 2021), 3D dense correspondence (Zhou et al., 2016), and super-resolution (Yuan et al., 2018). TrajectoryCrafter (Yu et al., 2025a) extends cycle consistency to multi-view synthesis through explicit forward-backward reprojection, requiring two full rendering passes. In contrast, our GRT reveals the analytical redundancy in this cycle and enables direct disocclusion mask derivation from target-view geometry alone.

## 3. Method

We present a self-supervised approach for training stereo inpainting networks using monocular videos (images). Building on the DIBR framework (Section 3.1), we identify data scarcity and domain gaps as key bottlenecks (Section 3.2). We introduce cycle consistency (Section 3.3) as a self-supervision mechanism and prove the Geometric Reciprocity Theorem (GRT) (Section 3.4), which eliminates the computational overhead of cycle consistency while preserving equivalent geometric constraints, enabling self-supervised learning from monocular videos (images) without requiring paired stereo data (Section 3.5).

### 3.1. Preliminaries

Stereoscopic video generation synthesizes stereo pairs from monocular input for immersive 3D visualization. Given a monocular frame as the left-eye view $I_L$, the task is to generate the corresponding right-eye view $I_R$. While the reverse direction is equally valid, we adopt the left-to-right formulation for clarity. Modern approaches employ the Depth-Image-Based Rendering (DIBR) framework, which reduces stereoscopic generation to a stereo inpainting problem. First, a depth estimation model $\mathcal{D}$ predicts disparity, which is inversely proportional to depth:

$$d_L = \mathcal{D}(I_L), \quad d_L = \frac{bf}{Z_L}, \tag{1}$$

where $d_L(x, y)$ denotes the disparity at pixel $(x, y)$, $Z_L(x, y)$ is the corresponding depth, $b$ is the baseline, and $f$ is the focal length. We adopt the convention that disparity values are positive, with larger values indicating closer objects. Second, warping projects the input to the target viewpoint. Specifically, $W_{L \rightarrow R}$ establishes pixel correspondences between views. For each pixel $(x, y)$ in the left view, its corresponding position $(x', y')$ in the right view is computed as:

$$x' = x - d_L(x, y), \quad y' = y, \tag{2}$$

where the horizontal coordinate is shifted by the disparity value, while the vertical coordinate remains unchanged due to rectified stereo geometry. During warping, multiple source pixels may map to the same target location (collision), or some target pixels may receive no mapping (disocclusion). The warping operation outputs the warped image $\tilde{I}_R$ and the disocclusion mask $M_{\text{dis}}^{L \rightarrow R}$:

$$\tilde{I}_R, M_{\text{dis}}^{L \rightarrow R} = W_{L \rightarrow R}(I_L, d_L), \tag{3}$$

where $M_{\text{dis}}^{L \rightarrow R}(x, y) = 1$ indicates pixels that are disoccluded in $\tilde{I}_R$ and require inpainting. Third, a stereo inpainting network fills these holes:

$$\hat{I}_R = G(\tilde{I}_R, M_{\text{dis}}^{L \rightarrow R}), \tag{4}$$

where $G$ represents the stereo inpainting network.

### 3.2. Motivation

The primary bottleneck in DIBR-based stereoscopic generation is training the stereo inpainting network $G$, stemming from the unique characteristics of stereoscopic disocclusion patterns and the scarcity of suitable training data.

**Domain gap in disocclusion patterns.** Stereoscopic disocclusion masks $M_{\text{dis}}^{L \rightarrow R}$ exhibit geometrically structured patterns fundamentally different from typical inpainting masks (random strokes, rectangular regions, or object instances). This domain gap causes general inpainting methods and training-free methods (Wang et al., 2024; Dai et al., 2024) to produce suboptimal results, lacking stereo-specific geometric priors for plausible disocclusion filling. **Data scarcity for supervised training.** Supervised approaches (Zhao et al., 2024; Huang et al., 2025) require training pairs $(I_R, M_{\text{dis}}^{L \rightarrow R})$, where the stereo inpainting network takes masked right view $I_R \odot (1 - M_{\text{dis}}^{L \rightarrow R})$ and disocclusion mask $M_{\text{dis}}^{L \rightarrow R}$ as input, using $I_R$ for supervision. However, no ready-to-use datasets exist. To construct such data from real stereo pairs, StereoCrafter (Zhao et al., 2024) collects real stereo video pairs and generates training data by aligning left to right views to identify disocclusion regions $M_{\text{dis}}^{L \rightarrow R}$. However, this alignment introduces stereo matching errors that severely degrade data quality. Moreover, high-quality stereo

## Cycle Consistency

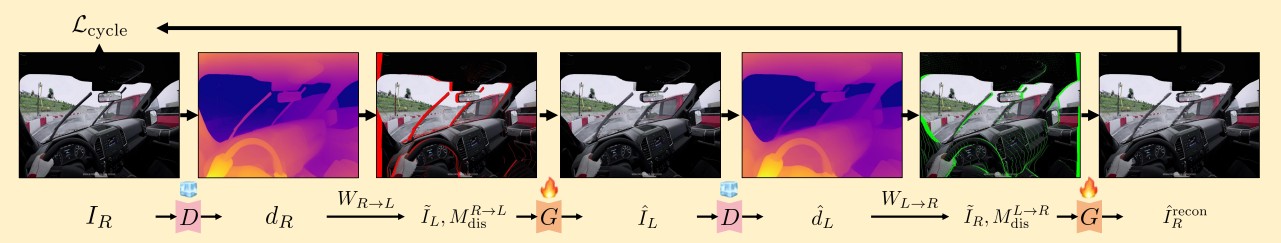

*Figure 1.* **Cycle consistency framework.** Given right view $I_R$, the complete cycle synthesizes left view through depth estimation ($\mathcal{D}$), forward warping ($W_{R \to L}$), and inpainting ($G$), then reconstructs the right view via depth estimation on the synthesized left view, backward warping ($W_{L \to R}$), and inpainting. The reconstruction loss $\mathcal{L}_{\text{cycle}} = \|\hat{I}_R^{\text{recon}} - I_R\|$ provides self-supervision.

videos predominantly come from copyrighted 3D films with restricted access, further limiting availability. Synthetic data approaches, such as Restereo (Huang et al., 2025) using Kubric (Greff et al., 2022), avoid stereo matching errors but suffer significant domain gaps between rendered and real-world videos, limiting real-world generalization.

To overcome these limitations, we propose the first self-supervised approach leveraging abundant monocular videos (images) for training stereo inpainting networks. Our method demonstrates that, based on cycle consistency, disocclusion masks $M_{\text{dis}}^{L \to R}$ can be directly derived from right views $I_R$ to construct training data, eliminating the need for stereo pairs or synthetic data.

### 3.3. Cycle Consistency

We introduce right-left-right cycle consistency as a self-supervision mechanism for stereoscopic generation. As illustrated in Figure 1, the principle is geometric and bidirectional: synthesizing the left view from the right, then reconstructing the right from this synthesized left, should recover the original input. Given a frame $I_R$, we first synthesize a pseudo left-eye view through the DIBR framework:

$$d_R = \mathcal{D}(I_R), \tag{5}$$

$$\tilde{I}_L, M_{\text{dis}}^{R \to L} = W_{R \to L}(I_R, d_R), \tag{6}$$

$$\hat{I}_L = G(\tilde{I}_L, M_{\text{dis}}^{R \to L}). \tag{7}$$

The right view is then reconstructed from $\hat{I}_L$:

$$\hat{d}_L = \mathcal{D}(\hat{I}_L), \tag{8}$$

$$\tilde{I}_R, M_{\text{dis}}^{L \to R} = W_{L \to R}(\hat{I}_L, \hat{d}_L), \tag{9}$$

$$\hat{I}_R^{\text{recon}} = G(\tilde{I}_R, M_{\text{dis}}^{L \to R}). \tag{10}$$

The cycle consistency loss enforces that reconstruction matches input:

$$\mathcal{L}_{\text{cycle}} = \|\hat{I}_R^{\text{recon}} - I_R\|. \tag{11}$$

While cycle consistency enables self-supervised training without paired stereo data, naive implementation is computationally prohibitive. End-to-end differentiability requires backpropagating through warping operations $W(\cdot, \cdot)$ that depend on estimated disparity $\mathcal{D}(\cdot)$, yet warping involves discrete pixel coordinate mapping and scattered writes to irregular locations, operations fundamentally non-differentiable in nature. Differentiable rendering approximations introduce substantial overhead and complexity. More critically, the cycle demands four sequential model inferences per training step: disparity estimation from $I_R$, left view inpainting, disparity re-estimation from $\hat{I}_L$, and final right view inpainting. This doubles memory requirements and training costs while accumulating errors through multiple non-linear transformations, making video-scale training computationally infeasible.

### 3.4. Geometric Reciprocity Theorem

We resolve all aforementioned challenges through a key geometric insight: the stereo warping process itself inherently encodes all information needed to enforce cycle consistency. Building on this insight, we mathematically prove that the entire gradient-intensive left view synthesis steps, including inpainting the left view, estimating its depth, and performing pixel warping operations, do not affect the cycle consistency loss $\mathcal{L}_{\text{cycle}}$ and can therefore be eliminated entirely. We formalize this result as follows:

**Geometric Reciprocity Theorem (GRT).** Under the nearest-neighbor DIBR formulation, for any right (target) view $I_R$ synthesized from a left (source) view, the disocclusion mask required for synthesis is mathematically equivalent to the mask of pixels that would be lost when warping from right (target) to left (source):

$$M_{\text{dis}}^{L \to R} = M_{\text{lost}}^{R \to L}, \tag{12}$$

where $M_{\text{lost}}^{R \to L}$ identifies pixels in $I_R$ that would be lost due to boundary violations or depth occlusions during right-to-left warping with estimated disparity $d_R = \mathcal{D}(I_R)$. By

## From Cycle Consistency to Geometric Reciprocity Theorem

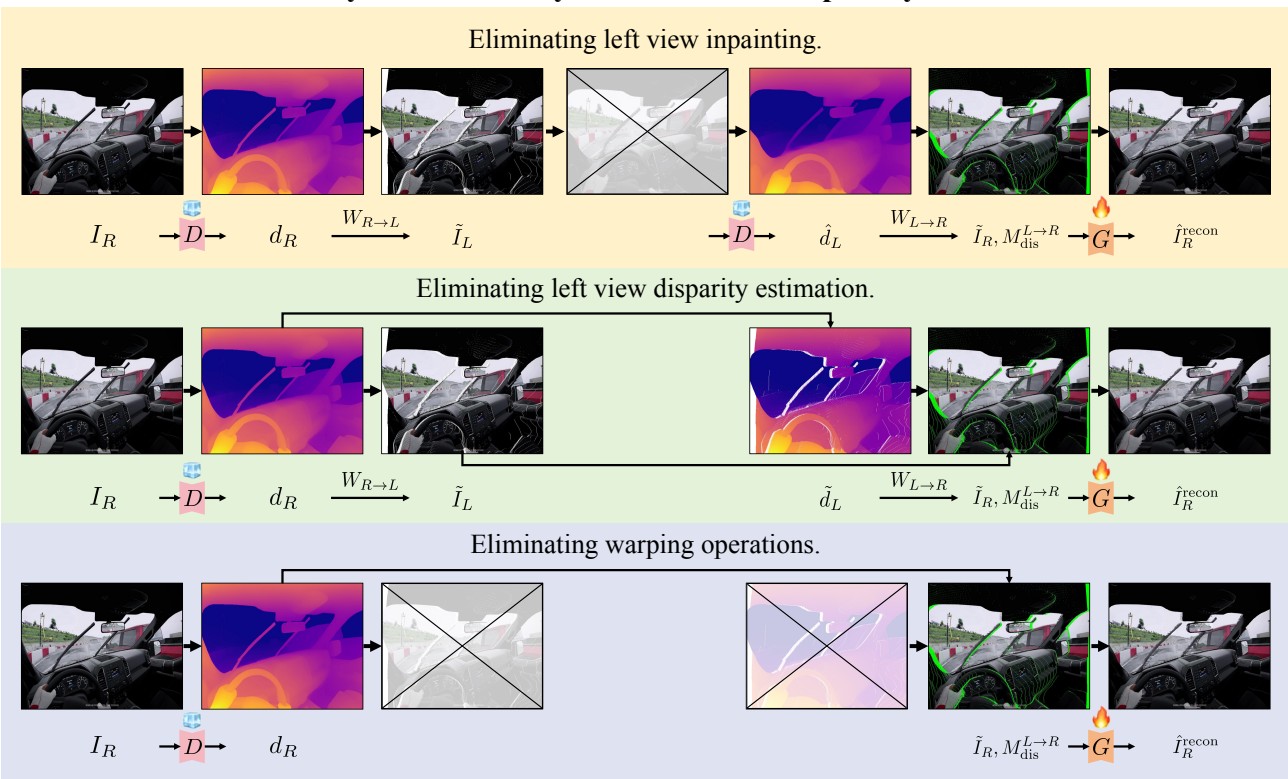

*Figure 2.* **Progressive simplification of cycle consistency to Geometric Reciprocity. (i)** Inpainted regions in $\hat{I}_L$ do not affect $\hat{I}_R^{\text{recon}}$, allowing us to skip left view inpainting (marked with $\times$) and directly use $\tilde{I}_L$. **(ii)** Right-to-left warping transfers disparity from $d_R$ to $\tilde{I}_L$, allowing us to skip left view disparity estimation and directly reuse $\tilde{d}_L$. **(iii)** Transferred disparity ensures perfect round-trips for all validly warped pixels, enabling analytical computation of $M_{\text{dis}}^{L \to R}$ as pixels lost during right-to-left warping and eliminating all warping operations (marked with $\times$). The final result reveals that $M_{\text{dis}}^{L \to R} = M_{\text{lost}}^{R \to L}$ can be computed directly from $(I_R, d_R)$ alone.

*treating any monocular image as a right (target) view, GRT enables us to directly compute the inference-time disocclusion mask that would emerge when synthesizing it from a left (source) view, using only depth estimation from $I_R$.* The image itself then serves as ground truth for these disocclusions, enabling self-supervised training from monocular videos (images) while bypassing the entire synthesis pipeline.

### 3.4.1. PROOF OF GRT

We prove the GRT by progressively simplifying the cycle consistency framework through three geometric observations visualized in Figure 2. Remarkably, we demonstrate that each computational step in the naive cycle can be eliminated without affecting the final result, revealing an elegant mathematical structure underlying stereoscopic generation. Note that this proof assumes nearest-neighbor warping; we extend to soft interpolation warping in Section A9.

**Eliminating left view inpainting.** We first observe that the inpainted left view content, $\hat{I}_L = G(\tilde{I}_L, M_{\text{dis}}^{R \to L})$, is geometrically irrelevant to the cycle consistency loss. Consider

pixels where $M_{\text{dis}}^{R \to L} = 1$: these regions were disoccluded during right-to-left warping and thus have no physical counterpart in the original $I_R$. When warping back from left to right, these inpainted pixels cannot produce valid mappings since they represent content absent from the actual scene geometry. Consequently, the disocclusion mask $M_{\text{dis}}^{L \to R}$ computed from the incomplete warped view $\tilde{I}_L$ is identical to that computed from the fully inpainted $\hat{I}_L$. This geometric property allows us to bypass the entire left view inpainting step:

$$\tilde{I}_R, M_{\text{dis}}^{L \to R} = W_{L \to R}(\tilde{I}_L, \mathcal{D}(\tilde{I}_L)). \tag{13}$$

**Eliminating left view disparity estimation.** Having established that we can work directly with $\tilde{I}_L$ in the previous step, we recognize that disparity estimation from the left view is redundant. During forward warping, each pixel $(x_R, y_R)$ in $I_R$ projects to position $(x_L, y_L)$ where:

$$x_L = x_R + d_R(x_R, y_R), \quad y_L = y_R. \tag{14}$$

Treating disparity as a single-channel image, this projection simultaneously transfers both color and disparity to non-

disoccluded pixels:

$$\tilde{I}_L(x_L, y_L) = I_R(x_R, y_R),$$
$$\tilde{d}_L(x_L, y_L) = d_R(x_R, y_R),$$
(15)

where $(x_L, y_L)$ receives projection from $(x_R, y_R)$. Note that $\tilde{d}_L$ contains valid disparity values only at non-disoccluded pixels transferred from $I_R$, while disoccluded regions (where $M_{\text{dis}}^{R \to L} = 1$) remain undefined. The crucial observation is that the previous step eliminated the need for complete left view information. We only need disparity values for non-disoccluded pixels in $\tilde{I}_L$, precisely those already transferred from $I_R$. Disoccluded regions do not participate in backward warping and thus do not require disparity estimation. Therefore, we directly use the transferred disparity $\tilde{d}_L$ instead of re-estimating via $\mathcal{D}(\tilde{I}_L)$, eliminating another computational bottleneck:

$$\tilde{I}_R, M_{\text{dis}}^{L \to R} = W_{L \to R}(\tilde{I}_L, \tilde{d}_L).$$
(16)

**Eliminating warping operations.** Our final observation is that even the warping operations themselves can be eliminated. After removing both left view inpainting and disparity estimation, the cycle reduces to forward-backward warping: estimate disparity $d_R$ from $I_R$, forward warp to obtain $\tilde{I}_L$ and $\tilde{d}_L$, then backward warp to compute $\tilde{I}_R$ and $M_{\text{dis}}^{L \to R}$. We observe that pixels completing the round-trip warping return to their exact original positions:

$$\begin{aligned} x_R' &= x_L - \tilde{d}_L(x_L, y_L) \\ &= [x_R + d_R(x_R, y_R)] - [d_R(x_R, y_R)] \quad (17) \\ &= x_R. \end{aligned}$$

This reveals the fundamental structure: pixels completing the forward-backward cycle remain unchanged, while those failing to complete it form the disocclusion mask. From a self-supervised learning perspective, this structure naturally provides supervision. Pixels completing the cycle correspond to regions where ground truth exists in $I_R$, while $M_{\text{dis}}^{L \to R}$ identifies regions requiring inpainting. Rather than explicitly performing warping operations to identify these regions, we determine which pixels from $I_R$ fail to complete the round-trip. A pixel $(x_R, y_R)$ is lost under two mutually exclusive conditions:

**(a) Boundary violation:** The pixel projects outside the image domain during forward warping:

$$M_{\text{oob}}^{R \to L}(x_R, y_R) = \mathbb{I}[x_R + d_R(x_R, y_R) \notin [0, W]], \quad (18)$$

where $W$ is image width and $\mathbb{I}[\cdot]$ is the indicator function.

**(b) Depth occlusion:** Multiple pixels from $I_R$ map to the same location $(x_L, y_L)$ during forward warping. Only the closest pixel is retained using a depth buffer:

$$B[x_L, y_L] = \max_{(x_R, y_R) \to (x_L, y_L)} d_R(x_R, y_R), \quad (19)$$

where $(x_R, y_R) \to (x_L, y_L)$ denotes all pixels from $I_R$ projecting to $(x_L, y_L)$ with $y_R = y_L$. A pixel is occluded if its disparity is less than this maximum:

$$M_{\text{occl}}^{R \to L}(x_R, y_R) = \mathbb{I}[d_R(x_R, y_R) < B[x_L, y_L]], \quad (20)$$

where $(x_L, y_L)$ is the projected location from $(x_R, y_R)$. The complete lost pixel mask combines both cases:

$$M_{\text{lost}}^{R \to L} = M_{\text{oob}}^{R \to L} \vee M_{\text{occl}}^{R \to L}. \quad (21)$$

Both conditions depend only on $(I_R, d_R)$ and require no warping operations, only analytical computation. This remarkable result establishes the GRT: by eliminating all three computational steps (left view inpainting, left view disparity estimation, and warping operations), we arrive at the elegant equivalence $M_{\text{dis}}^{L \to R} = M_{\text{lost}}^{R \to L}$. Note that if we start from the left-right-left cycle instead, we can derive the symmetric result $M_{\text{dis}}^{R \to L} = M_{\text{lost}}^{L \to R}$.

### 3.4.2. EQUIVALENCE TO FULL SUPERVISION

Unlike most self-supervised methods (Zhu et al., 2017; Yang et al., 2021) that provide weak, indirect supervision for downstream tasks, GRT fundamentally differs by producing the same mask supervision as paired stereo data under the stated DIBR formulation. By treating any monocular image as a target view $I_R$, GRT directly computes $M_{\text{dis}}^{L \to R} = M_{\text{lost}}^{R \to L}$ from disparity estimation alone. **This is the same mask that emerges when a source view synthesizes $I_R$ via DIBR (Figure 3, top)**, yet requires neither source view synthesis nor paired stereo data. As shown in the green-highlighted regions of Figure 3, the mask-conditioned inpainting process during training matches that at inference, ensuring train-inference consistency.

### 3.4.3. APPLICATION OF GRT

GRT enables self-supervised training data construction by computing inference-identical disocclusion masks from monocular images (Figure 3, bottom). Each mask can be computed once offline and reused during training. Given a target view $I_R$, we estimate $d_R = \mathcal{D}(I_R)$ and compute $M_{\text{dis}}^{L \to R} = M_{\text{lost}}^{R \to L}$ to construct triplets $(X_{\text{input}}, M_{\text{dis}}^{L \to R}, I_R)$ where $X_{\text{input}} = I_R \odot (1 - M_{\text{dis}}^{L \to R})$ with $I_R$ as ground truth. The training objective is:

$$\mathcal{L} = \|G(X_{\text{input}}, M_{\text{dis}}^{L \to R}) - I_R\|. \quad (22)$$

**Training and evaluation datasets.** We construct the first comprehensive datasets for stereo inpainting from widely-adopted benchmarks: ImageNet-GRT and Kinetics-GRT from ImageNet (Russakovsky et al., 2015) and Kinetics (Kay et al., 2017) for training. Building on the established equivalence (Section 3.4.2), we create DAVIS-GRT from DAVIS (Perazzi et al., 2016) for evaluation (green-highlighted regions, Figure 3) to faithfully reflect real-world inference performance (details in Section A2).

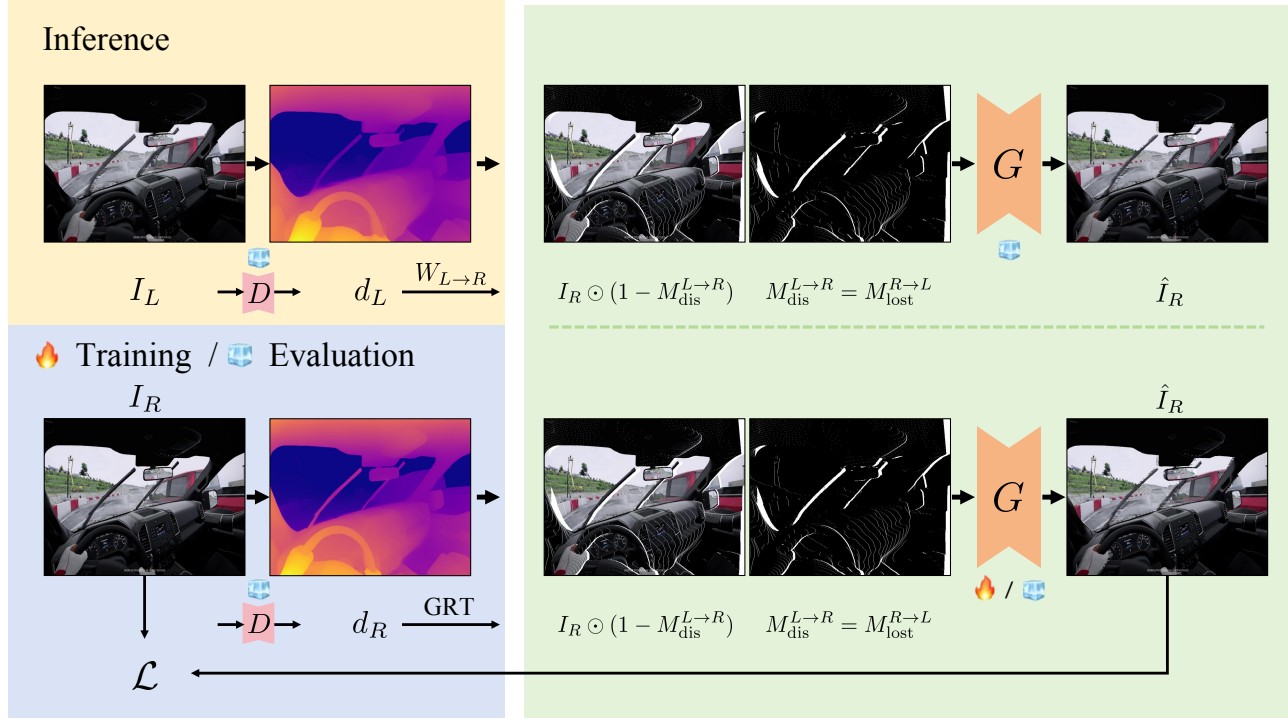

*Figure 3.* **Equivalence to Full Supervision. Top:** inference synthesizes $I_R$ from $I_L$ via DIBR and obtains $M_{\mathrm{dis}}^{L \to R}$ for inpainting. **Bottom:** GRT treats a monocular frame as $I_R$ and computes the same mask analytically from $I_R$ alone for training and DAVIS-GRT evaluation.

## 3.5. Model Architecture

We adopt LaMa (Suvorov et al., 2022), which leverages Fast Fourier Convolutions, for image stereo inpainting and ProPainter (Zhou et al., 2023) for video stereo inpainting. ProPainter features recurrent flow completion and sparse video transformers to handle temporal consistency. Both models are fine-tuned on ImageNet-GRT and Kinetics-GRT respectively, using a combined loss of L1 and perceptual components.

## 4. Experiments

### 4.1. Experimental Setup

**Baselines.** For image stereo inpainting, we compare against StereoDiffusion (Wang et al., 2024), ZeroStereo (Wang et al., 2025), and Mono2Stereo (Yu et al., 2025b). For video stereoscopic generation, we evaluate against SVG (Dai et al., 2024) and StereoCrafter (Zhao et al., 2024). We denote our methods as **Ours-Image** and **Ours-Video**.

**Datasets and Metrics.** We conduct evaluations on two complementary benchmarks. DAVIS-GRT isolates the stereo inpainting sub-task under geometrically consistent masks with clean ground truth, while Inria 3DMovie evaluates the full left-to-right stereoscopic generation pipeline on real stereo footage. On DAVIS-GRT, which comprises 50 video

clips with 3,455 frames, we report PSNR, SSIM (Wang et al., 2004), and LPIPS (Zhang et al., 2018) to assess stereo inpainting quality, along with CLIP Temporal Consistency (CTC), which measures frame-to-frame CLIP feature cosine similarity, to evaluate temporal coherence. Image-based methods are evaluated per-frame on this dataset. Per-frame inference time is reported at $512 \times 512$ resolution. On the Inria 3DMovie Dataset (Alahari et al., 2013), which contains 36 stereo video pairs, we evaluate video stereo generation methods using SIoU (Yu et al., 2025b) and MEt3R (Asim et al., 2025) to measure viewing comfort and geometric consistency, reflecting the quality of stereoscopic viewing experience.

### 4.2. Quantitative Results

**DAVIS-GRT.** As shown in Table 2, both Ours-Image and Ours-Video achieve superior performance across all metrics compared to existing methods, while being significantly faster with inference times of 0.05s and 0.24s per frame, respectively. This improvement stems from our self-supervised training paradigm that ensures train-inference alignment through GRT-derived data, enabling more accurate geometric understanding without relying on imperfect stereo pair collections.

**Inria 3DMovie Dataset.** Leveraging improved stereo in-

*Table 2.* DAVIS-GRT stereo inpainting results. Video methods additionally report temporal consistency (CTC).

| Method | PSNR↑ | SSIM↑ | LPIPS↓ | CTC↑ | Time (s)↓ |
|---|---|---|---|---|---|
| *Image Stereo Inpainting* | | | | | |
| Mono2Stereo | 28.20 | 0.9250 | 0.0485 | – | 2.4 |
| ZeroStereo | 28.50 | 0.9280 | 0.0470 | – | 4.1 |
| StereoDiffusion | 29.77 | 0.9360 | 0.0411 | – | 10.4 |
| **Ours-Image** | **35.52** | **0.9800** | **0.0129** | – | **0.05** |
| *Video Stereo Inpainting* | | | | | |
| SVG | 27.33 | 0.9052 | 0.0552 | 0.9721 | 3.0 |
| StereoCrafter | 28.95 | 0.9501 | 0.0445 | 0.9755 | 0.6 |
| **Ours-Video** | **34.06** | **0.9733** | **0.0210** | **0.9770** | **0.24** |

painting capabilities, our methods deliver more comfortable viewing experiences. As shown in Table 3, Ours-Video demonstrates substantially superior geometric consistency and viewing comfort compared to video stereoscopic generation baselines.

*Table 3.* Evaluation on Inria 3DMovie.

| Method | SIoU↑ | MEt3R↓ |
|---|---|---|
| SVG | 0.2160 | 0.2245 |
| StereoCrafter | 0.2201 | 0.1961 |
| **Ours-Video** | **0.2516** | **0.0973** |

### 4.3. Qualitative Results

Visual comparisons in Figure 4 demonstrate the effectiveness of our approach. We compare against StereoDiffusion, the top-performing image baseline, as well as all video stereo inpainting methods. Our GRT-based training enables precise geometric understanding, producing natural textures in disoccluded regions with smooth boundary transitions.

### 4.4. Effectiveness of GRT Training Data

To validate the broad applicability of GRT-derived training data, we fine-tune several foundation models using our datasets. For images, we adapt LaMa (Suvorov et al., 2022) and pretrained Stable Diffusion (Rombach et al., 2022) inpainting models (SD1.5, SD2.1, and SDXL); for videos, we fine-tune ProPainter (Zhou et al., 2023) and StereoCrafter. As shown in Table 4, all models achieve substantial improvements across metrics after fine-tuning on GRT data. Notably, even StereoCrafter, purpose-built for stereoscopic generation, benefits considerably, demonstrating that GRT provides higher-quality supervision than existing stereo pair collection strategies. These results confirm our self-supervised training paradigm is broadly applicable across diverse architectures.

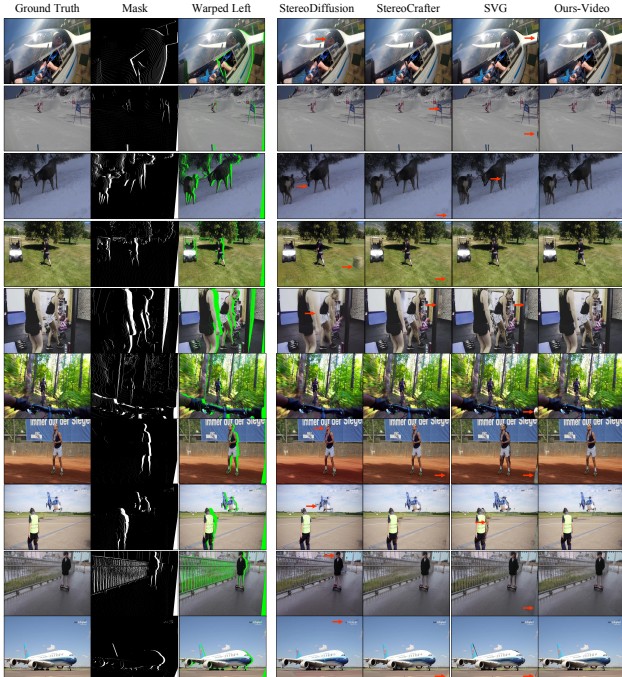

*Figure 4.* **Qualitative comparison.** Our method produces more natural textures and smoother boundaries.

*Table 4.* GRT fine-tuning results across image and video inpainting backbones.

| Method | PSNR↑ | SSIM↑ | LPIPS↓ | CTC↑ |
|---|---|---|---|---|
| *Image Stereo Inpainting* | | | | |
| LaMa | 31.75 | 0.9660 | 0.0239 | – |
| LaMa + GRT | **35.52** | **0.9800** | **0.0129** | – |
| SD1.5 Inpainting | 27.38 | 0.9152 | 0.0982 | – |
| SD1.5 Inpainting + GRT | **30.70** | **0.9574** | **0.0281** | – |
| SD2.1 Inpainting | 27.37 | 0.9143 | 0.0981 | – |
| SD2.1 Inpainting + GRT | **28.12** | **0.9472** | **0.0419** | – |
| SDXL Inpainting | 23.40 | 0.8842 | 0.1426 | – |
| SDXL Inpainting + GRT | **29.45** | **0.9431** | **0.0481** | – |
| *Video Stereo Inpainting* | | | | |
| ProPainter | 31.03 | 0.9648 | 0.0318 | 0.9764 |
| ProPainter + GRT | **34.06** | **0.9733** | **0.0210** | **0.9770** |
| StereoCrafter | 28.95 | 0.9501 | 0.0445 | 0.9755 |
| StereoCrafter + GRT | **30.85** | **0.9612** | **0.0298** | **0.9760** |

# 5. Conclusion

We present a self-supervised framework for training stereo inpainting networks from monocular videos, addressing the data bottleneck in monocular-to-stereo conversion. Our key contribution is the Geometric Reciprocity Theorem (GRT): under the stated DIBR formulation, the disocclusion mask for a target view equals the pixels lost when warping back to source. This enables computing training masks from depth alone, eliminating stereo pairs or synthetic data. We achieve state-of-the-art quality with scalable training from real-world videos.

## Acknowledgements

This work is supported by Hong Kong Research Grants Council – General Research Fund (Grant Nos. 17213825 and 17211024), Hong Kong Innovation and Technology Commission – Innovation and Technology Fund (Project No. ITS/488/24FP), and HKU Seed Fund for PI Research.

## Impact Statement

This paper presents work whose goal is to advance the field of machine learning. There are many potential societal consequences of our work, none of which we feel must be specifically highlighted here.

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

## A1. Code and Data Release

The project repository is `https://github.com/Visual-AI/GRT`. We will release source code, precomputed GRT masks, and trained weights under the Apache 2.0 license.

## A2. Dataset Details

We construct three datasets by applying the Geometric Reciprocity Theorem (GRT) (Section 3.4) to widely-adopted vision benchmarks. For each image or video frame, we treat it as a right (target) view, estimate its monocular depth, and compute a binary disocclusion mask $M_{\text{dis}}^{L \to R}$ following the GRT procedure. This yields self-supervised training samples and test-aligned evaluation samples for stereo inpainting, where each RGB image or frame is paired with its corresponding GRT-derived disocclusion mask.

**Depth estimation and GRT mask generation.** We use Depth Anything V2-Large (Yang et al., 2024) to estimate monocular depth for each image or frame, following previous stereoscopic video generation literature. The predicted relative (inverse) depth is converted to a disparity map and linearly rescaled to $[0, \alpha W]$, where $W$ is the image width and $\alpha = 0.1$ for training and $\alpha = 0.06$ for evaluation. We then apply the GRT procedure (Section 3.4) to compute $M_{\text{dis}}^{L \to R}$ by identifying lost pixels through boundary violations and depth occlusions. On average, disoccluded regions comprise approximately 10% of pixels for training data. Note that our framework flexibly accommodates alternative depth estimators, such as Video Depth Anything (Chen et al., 2025) for temporally consistent depth, Depth Anything V3 (Lin et al., 2025) for metric depth, or DepthFocus (Min et al., 2025) for multi-layer depth to handle transparent surfaces. We adopt Depth Anything V2 to follow prior works and ensure fair comparisons.

**ImageNet-GRT.** ImageNet-GRT uses the complete ImageNet-1K training split (Russakovsky et al., 2015), containing ~1.28M images across 1,000 object categories. We utilize all images without filtering and disregard class labels, treating ImageNet purely as a diverse source of natural imagery.

**Kinetics-GRT.** Kinetics-GRT uses the Kinetics-400 training split (Kay et al., 2017), comprising ~240K videos across 400 human action categories with rich temporal dynamics and camera motion. We decode all frames at their original resolution and frame rate while preserving temporal order.

**DAVIS-GRT.** DAVIS-GRT uses all 50 videos (3,455 frames) from DAVIS 2016 (Perazzi et al., 2016), a video segmentation benchmark known for high-quality annotations and challenging scenarios. DAVIS-GRT serves *exclusively* for testing; no DAVIS frames are used during training or model selection, ensuring unbiased assessment of generalization.

## A3. Pseudocode

We provide pseudocode in Figure A1 implementing the Geometric Reciprocity Theorem (GRT) presented in Section 3.4. Given a disparity map, the function computes the disocclusion mask by identifying pixels that would be lost during warping due to boundary violations or depth occlusions.

## A4. Model Architecture and Training

Our model builds upon the LaMa architecture (Suvorov et al., 2022) with Fast Fourier Convolutions (FFC) for image stereo inpainting, and ProPainter (Zhou et al., 2023) for video stereo inpainting.

### A4.1. Image Stereo Inpainting

**Architecture.** We use the Big LaMa-Fourier generator (27M parameters): 3 downsampling blocks, 9 FFC-based residual blocks, and 3 upsampling blocks. The network takes 4-channel input (masked RGB + mask) and outputs 3-channel RGB. FFC blocks provide image-wide receptive fields by processing features in both spatial (local) and frequency (global) domains.

**Training.** We fine-tune the model on ImageNet-GRT with realistic disocclusion masks generated via GRT for 30 epochs at $256 \times 256$ resolution. We use combined L1 and LPIPS loss with Adam optimizer (learning rate $10^{-4}$), batch size 32, and cosine annealing scheduler on two NVIDIA V100 GPUs.

### A4.2. Video Stereo Inpainting

**Architecture.** We adopt ProPainter which comprises three key components: (1) Recurrent Flow Completion (RFC) network that completes corrupted optical flows using deformable alignment with $8\times$ downsampled features; (2) Dual-domain propagation combining global image propagation (with flow consistency check) and local feature propagation (flow-guided deformable alignment); (3) Mask-guided sparse video Transformer with 8 blocks, window size $5 \times 9$, and extended size equal to half of the window size. The Transformer applies sparse attention only to query windows intersecting mask regions and uses temporal stride of 2 for key/value space to reduce redundancy.

**Training.** We fine-tune from pretrained ProPainter weights on Kinetics-GRT for 10,000 steps at $432 \times 240$ resolution. We use combined L1 reconstruction loss and T-PatchGAN adversarial loss (weight 0.01) with Adam optimizer (learning rate $10^{-4}$) and batch size 8. The model processes local sequences of length 10 during training and length 20 during inference on two NVIDIA V100 GPUs.

```
1   def compute_grt_mask(disparity, direction='R2L'):
2       """
3       Compute GRT-based disocclusion mask via geometric warping.
4
5       Args:
6           disparity: [B,H,W] disparity map (positive values)
7           direction: 'R2L' for right-to-left warp (finds L->R disocclusions)
8                      'L2R' for left-to-right warp (finds R->L disocclusions)
9       Returns:
10          mask: [B,H,W] binary mask (True = disoccluded pixel)
11      """
12      B, H, W = disparity.shape
13
14      # Compute target positions after warping
15      x = torch.arange(W)[None, None, :]  # [1,1,W]
16      if direction == 'R2L':
17          x_target = round(x + disparity)  # Right-to-left warp
18      else:  # L2R
19          x_target = round(x - disparity)  # Left-to-right warp
20
21      # Mask 1: Out-of-bounds pixels (boundary violations)
22      mask_oob = (x_target < 0) | (x_target >= W)
23
24      # Mask 2: Occluded pixels (depth ordering conflicts)
25      # When multiple pixels map to same target, keep only the closest (max disparity)
26      valid_idx = torch.where(~mask_oob)
27      x_tgt = x_target[valid_idx]
28      d_src = disparity[valid_idx]
29
30      # Find winning disparity at each target position
31      d_winner = scatter_max(d_src, index=x_tgt, dim_size=W)  # [B,H,W]
32
33      # Mark projected pixels that lost the depth competition
34      mask_occ = torch.zeros_like(mask_oob)
35      is_occluded = (d_src < d_winner[valid_idx])
36      mask_occ[valid_idx][is_occluded] = True
37
38      # Combine both invalidation conditions
39      return mask_oob | mask_occ
```

*Figure A1.* **Pseudocode implementing the Geometric Reciprocity Theorem (GRT).** Given a disparity map, the function computes the disocclusion mask by identifying pixels lost during warping due to boundary violations (mask_oob) or depth occlusions (mask_occ).

## A5. Additional Analyses

**Naive cycle consistency.** GRT precomputes masks offline and reduces online optimization to standard inpainting, while naive cycle consistency requires multiple sequential model inferences and gradient approximations through warping. A controlled LaMa comparison is summarized in Table A1.

*Table A1.* Efficiency and quality comparison with naive cycle consistency.

| Method | Steps/s↑ | Memory↓ | PSNR↑ | SSIM↑ | LPIPS↓ |
|---|---|---|---|---|---|
| Naive Cycle Consistency | 0.6 | 19 GB | 32.14 | 0.9663 | 0.0271 |
| GRT | **1.9** | **11 GB** | **33.10** | **0.9718** | **0.0232** |

**Mask agreement.** We compare analytically computed GRT masks with cycle-consistency masks under different depth and warping settings. The high agreement in Table A2 indicates that the derived masks capture the same disocclusion structure in practice.

*Table A2.* Pixel-level mask agreement with GRT masks.

| Setting | Agreement (%)↑ |
|---|---|
| Cycle consistency, depth-consistent | 100.0 |
| Cycle consistency, independently estimated | 99.1 |
| Bilinear warping vs. nearest-neighbor GRT | 99.5 |

**Depth estimator robustness.** GRT is depth-estimator-agnostic and only requires a disparity map as input. Table A3 shows that performance degrades gradually with weaker depth backbones but remains close.

*Table A3.* Depth estimator sensitivity on Inria 3DMovie.

| Depth Estimator | SIoU↑ | MEt3R↓ |
|---|---|---|
| Depth Anything V2-Large | **0.2516** | **0.0973** |
| Depth Anything V2-Base | 0.2489 | 0.1012 |
| Depth Anything V1-Small | 0.2451 | 0.1058 |

**GRT mask components.** The boundary-violation and depth-occlusion terms are jointly required to identify geometrically lost pixels. Ablating either component degrades downstream inpainting quality.

**Data scaling.** Because GRT requires no paired stereo capture, it can benefit from larger monocular video corpora. Table A5 shows monotonic gains as the training set grows.

## A6. Limitations

GRT inherits the assumptions of DIBR-based stereoscopic generation. It relies on rectified stereo geometry and the quality of the input disparity map; transparent or reflective surfaces, inaccurate depth discontinuities, and non-Lambertian effects can therefore lead to incorrect masks

*Table A4.* Ablation of GRT mask components on LaMa trained with ImageNet-GRT.

| Mask Configuration | PSNR↑ | SSIM↑ | LPIPS↓ |
|---|---|---|---|
| Boundary violation only | 32.89 | 0.9694 | 0.0200 |
| Depth occlusion only | 32.54 | 0.9676 | 0.0216 |
| Full GRT mask | **35.52** | **0.9800** | **0.0129** |

*Table A5.* Dataset scale ablation for ProPainter trained on GRT data.

| Training Data Scale | PSNR↑ | SSIM↑ | LPIPS↓ | CTC↑ |
|---|---|---|---|---|
| 0% (pretrained) | 31.03 | 0.9648 | 0.0318 | 0.9764 |
| 10% (∼24K videos) | 31.88 | 0.9672 | 0.0274 | 0.9765 |
| 50% (∼120K videos) | 33.45 | 0.9710 | 0.0241 | 0.9768 |
| 100% (∼240K videos) | 34.06 | 0.9733 | 0.0210 | 0.9770 |
| Kinetics-700 (∼650K videos) | **34.83** | **0.9751** | **0.0188** | **0.9774** |

or visually implausible inpainting. The nearest-neighbor formulation gives exact mask consistency under the theorem statement, while bilinear interpolation introduces a small approximation gap as discussed in Section A9.

## A7. Discussion of General Inpainting Models

General-purpose image inpainting models are designed to fill large contiguous missing regions by hallucinating plausible content from surrounding context. While these models are powerful, stereo disocclusions present a fundamentally different mask pattern. Disocclusion masks are thin, elongated, and scattered along depth discontinuities at object boundaries, with significantly smaller total area compared to typical inpainting masks. More critically, disocclusions reveal previously hidden background content that is often visually and semantically distinct from the adjacent occluding foreground. For example, a disocclusion behind a person might expose a wall or furniture with no visual similarity to the person's appearance. This domain gap in both mask geometry and content semantics causes general inpainting models to produce inferior results on stereo disocclusions. As we demonstrate in Figure A2, while these models succeed on large contiguous masks, they fail on the thin scattered masks characteristic of stereo disocclusions.

## A8. Training Data Challenges

Constructing high-quality training data remains a significant bottleneck for stereo inpainting. Methods relying on real stereo pairs, such as StereoCrafter (Zhao et al., 2024), face a fundamental paradox: they rely on stereo matching algorithms to extract pixel correspondences from stereo pairs and subsequently derive disocclusion masks, yet stereo matching itself is prone to errors and particularly unreliable in disoccluded regions. As shown in Fig. A3, stereo matching produces distorted masks with alignment errors.

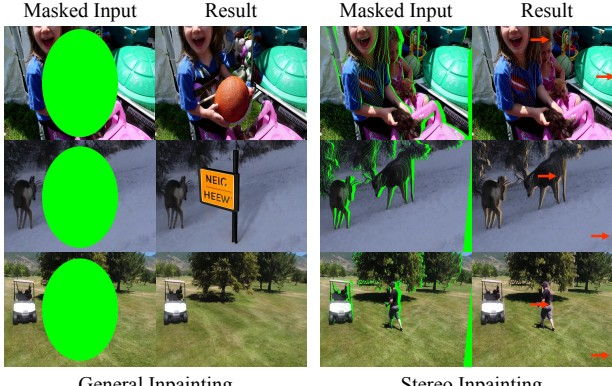

Masked Input    Result    Masked Input    Result

General Inpainting          Stereo Inpainting

*Figure A2.* Performance of SDXL Inpainting on different mask patterns. The model handles large contiguous masks well (General Inpainting) but struggles with thin scattered disocclusion masks along object boundaries (Stereo Inpainting).

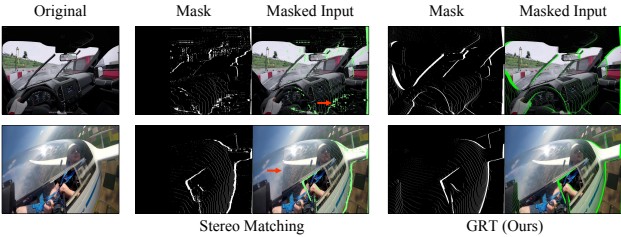

Original    Mask    Masked Input    Mask    Masked Input

Stereo Matching          GRT (Ours)

*Figure A3.* Stereo matching methods produce distorted disocclusion masks and misaligned inpainting triplets. Our GRT approach generates geometrically accurate masks at test-time.

StereoCrafter confirms this limitation, acknowledging the prevalence of **large stereo matching errors** that force them to employ complex manual alignment and aggressive filtering strategies (e.g., discarding samples where warping PSNR $< 25$ dB), resulting in substantial data waste. Alternatively, approaches utilizing synthetic data (Huang et al., 2025) provide precise geometry but suffer from inevitable **Sim2Real domain gaps**. Our GRT-based approach resolves these dilemmas by deriving geometrically consistent masks directly from monocular data, avoiding both error-prone matching and synthetic domain shifts.

## A9. Extension to Soft Interpolation Warping

The GRT proof in Section 3.4.1 assumes nearest-neighbor warping, where each pixel $(x_R, y_R)$ maps to discrete coordinates $(x_L, y_L) = (\text{round}(x_R + d_R(x_R, y_R)), y_R)$. In practice, bilinear warping is sometimes preferred for smoother synthesis. We show that GRT extends to this soft interpolation setting by reformulating the problem at the renderer level rather than the pixel level, though this introduces a subtle train-test consistency trade-off.

**Renderer-based formulation.** Under bilinear warping, we treat each pixel $(x_R, y_R)$ in the right view $I_R$ as a renderer that carries appearance and geometry information. Each

renderer projects to continuous coordinates:

$$x'_L = x_R + d_R(x_R, y_R), \quad y'_L = y_R, \tag{23}$$

and distributes its contribution to a $2 \times 2$ neighborhood in the left view via bilinear weights $w_{x_R, y_R}(x_L, y_L)$. This splatting process synthesizes the left view as a weighted blend of renderer contributions.

**Forward warping with occlusion handling.** For each left-view pixel $(x_L, y_L)$, let $S(x_L, y_L)$ denote all renderers whose bilinear kernels overlap it. To handle occlusions, we maintain a depth buffer that tracks the maximum disparity among contributing renderers:

$$d_L^{\max}(x_L, y_L) = \max_{(x_R, y_R) \in S(x_L, y_L)} d_R(x_R, y_R). \tag{24}$$

Only renderers at maximum disparity are retained, as nearer objects occlude farther ones. Let $S_{\max}(x_L, y_L)$ denote renderers at maximum disparity:

$$\begin{aligned} S_{\max}(x_L, y_L) = \{(x_R, y_R) \in S(x_L, y_L) : \\ d_R(x_R, y_R) = d_L^{\max}(x_L, y_L)\}. \end{aligned} \tag{25}$$

The warped left view is synthesized as:

$$\tilde{I}_L(x_L, y_L) = \frac{\sum_{(x_R, y_R) \in S_{\max}} w_{x_R, y_R} \cdot I_R(x_R, y_R)}{\sum_{(x_R, y_R) \in S_{\max}} w_{x_R, y_R}}, \tag{26}$$

where we abbreviate $S_{\max}(x_L, y_L)$ as $S_{\max}$ for notational simplicity.

**Lost renderer criteria.** A renderer $(x_R, y_R)$ is lost if it fails to contribute to any left-view pixel. This occurs under two conditions. First, boundary violation occurs when the renderer's bilinear kernel support lies entirely outside the valid image domain:

$$M_{\text{oob}}^{R \to L}(x_R, y_R) = \mathbb{I}\left[\lfloor x'_L \rfloor, \lceil x'_L \rceil \notin [-1, W]\right], \tag{27}$$

where the range $[-1, W]$ ensures at least one neighbor in $[0, W)$ has non-zero weight. Second, depth occlusion occurs when the renderer is occluded at all left-view locations in its support. Let $S^*(x_R, y_R)$ denote valid left-view pixels overlapping $(x_R, y_R)$'s bilinear kernel. The occlusion mask is:

$$\begin{aligned} M_{\text{occl}}^{R \to L}(x_R, y_R) = \mathbb{I}\big[d_R(x_R, y_R) < d_L^{\max}(x_L, y_L), \\ \forall (x_L, y_L) \in S^*(x_R, y_R)\big]. \end{aligned} \tag{28}$$

The complete lost mask combines both conditions:

$$M_{\text{lost}}^{R \to L} = M_{\text{oob}}^{R \to L} \vee M_{\text{occl}}^{R \to L}. \tag{29}$$

**GRT validity at the renderer level.** The Geometric Reciprocity relation $M_{\text{dis}}^{L \to R} = M_{\text{lost}}^{R \to L}$ continues to hold when

formulated at the renderer level. The three-step simplification in Section 3.4.1 remains geometrically valid: disoccluded regions lack scene correspondence, each renderer co-transfers disparity with its appearance, and renderers completing the round-trip return to their original positions. Therefore, lost renderers in the forward pass correspond exactly to disoccluded regions in the backward pass.

**Train-test consistency.** While GRT holds mathematically at the renderer level, bilinear warping introduces a subtle inconsistency. During training, $M_{\text{lost}}^{R \to L}$ is computed from discrete renderers in the original image $I_R$. During testing, the synthesized view $\hat{I}_L$ contains blended pixel values, and backward warping treats each blended pixel as a single renderer rather than the multiple discrete renderers that created it. This approximation causes slight differences in disocclusion masks, though the effect is minor for typical stereo baseline ranges.

**Implementation.** We adopt nearest-neighbor warping to maintain consistency with the theorem statement and to improve computational efficiency. Both variants produce visually similar results, while nearest-neighbor warping keeps training masks aligned with the stereoscopic generation pipeline.

