# OpenReview forum: "Geometric Reciprocity: Unlocking Self-Supervision for Stereoscopic Video Generation"
_ICML.cc/2026/Conference — ICML 2026 regular_

### Official Review · Reviewer_avs3 · 2026-03-04

**Soundness:** 3
**Presentation:** 3
**Significance:** 3
**Originality:** 3
**Overall Recommendation:** 4
**Confidence:** 3

**Summary:**

This paper introduces the Geometric Reciprocity Theorem (GRT) to train disocclusion inpainting for monocular-to-stereo conversion without paired stereo data. GRT analytically computes training masks from a single image and its depth map. The authors built large-scale self-supervised datasets, fine-tuned models, and achieved significant quality and speed gains across multiple backbones.

**Compliance With Llm Reviewing Policy:**

Affirmed.

**Final Justification:**

Thank you for the detailed rebuttal and additional clarifications. My main concerns have been adequately addressed.

In particular, I appreciate the added evidence on robustness to different depth estimators, the clearer explanation of the complementary roles of DAVIS-GRT and Inria 3DMovie, and the more explicit discussion of the method’s limitations and geometric assumptions. The clarification regarding the rationale for the train/test disparity scaling choice is also helpful, even though a full sensitivity sweep is deferred to future work.

Overall, the rebuttal makes the paper more complete and convincing.

**Key Questions For Authors:**

How do the results hold up if you swap depth estimators or inject controlled noise/bias into the disparity?

What is the principled rationale for your chosen $\alpha$ (especially train/test differences)? Can you provide sensitivity curves showing its impact on performance and mask statistics?

On synthetic/multi-view data with ground-truth disparity, can you quantify GRT mask accuracy against stereo-matching-derived masks (e.g., using IoU or Precision-Recall)?

**Limitations:**

The paper must explicitly discuss depth dependency, geometric assumptions, evaluation limitations on DAVIS-GRT, and standard data licensing/privacy disclaimers.

**Strengths And Weaknesses:**

Strengths

The GRT framework introduces a streamlined approach to self-supervised view synthesis by replacing cumbersome cycle-consistency constraints with an explicit, analytically pre-computable mask. This design ensures that generated training masks perfectly align with the disocclusion structures encountered during real-world inference.By simplifying the optimization objective, the method achieves substantial improvements in both processing speed and output quality across various backbones, as evidenced by results on the DAVIS-GRT and Inria 3DMovie datasets. Furthermore, its architecture presents a highly scalable, self-supervised paradigm that is modular enough to be integrated into and benefit other contemporary view-synthesis systems.

Weaknesses

Despite its efficiency, the paper lacks a rigorous error propagation analysis, specifically regarding how monocular depth inaccuracies or noise influence the final mask quality and stereo synthesis. The evaluation is somewhat self-referential, as the DAVIS-GRT benchmark relies on the same underlying assumptions as the GRT method itself; consequently, a broader validation against diverse, real-world stereo datasets is necessary to prove generalizability.Additionally, the quantitative assessment feels incomplete due to the exclusion of recent training-based video stereo SOTA models in the primary tables. Finally, the study would benefit from a more technical "deep dive" into its hyperparameters and architectural choices, such as the impact of disparity scaling ($\alpha$), specific warping interpolation methods, and the individual contributions of different mask components.

---

> ### Author Rebuttal · Authors · 2026-03-31
>
> ## Response to Reviewer avs3
>
> We thank the reviewer for the careful and constructive feedback. We address all concerns below, consolidating closely related points where appropriate.
>
> ---
>
> **W1 + Q1 + Q3: Depth Estimator Robustness and GRT Mask Quality**
>
> **Depth estimator sensitivity.** To quantify robustness to depth estimator choice, we evaluate stereo viewing quality on the Inria 3DMovie Dataset under different backbones with all other settings fixed:
>
> | Depth Estimator | SIoU↑ | MEt3R↓ |
> |---|---|---|
> | DepthAnythingV2-Large (default) | 0.2516 | 0.0973 |
> | DepthAnythingV2-Base | 0.2489 | 0.1012 |
> | DepthAnythingV1-Small | 0.2451 | 0.1058 |
>
> Performance degrades only modestly with weaker backbones. This is consistent with the observation that human stereo perception is relatively insensitive to fine-grained depth inaccuracies, provided that coarse boundary structure is preserved.
>
> **On GRT mask accuracy relative to stereo-matching-derived masks (Q3).** GRT is a mask derivation procedure that accepts any disparity map, **whether from a monocular estimator or a stereo matching pipeline**, and analytically outputs the disocclusion mask that a DIBR synthesis step would produce at inference time. Using a monocular estimator as input is motivated by two considerations: first, real stereo data is scarce and often inaccessible due to copyright restrictions; second, since inference-time masks are derived from monocular depth, using the same source during training ensures train-test mask consistency. We note that monocular estimators predict relative rather than metric depth, making a direct geometric comparison with metric stereo matching difficult. The independent validation on Inria 3DMovie (Table 3), a real-world stereo benchmark with no methodological overlap with GRT construction, provides the most direct evidence of generalizability.
>
> ---
>
> **W2: Evaluation on DAVIS-GRT**
>
> We appreciate the reviewer raising this concern and want to clarify the complementary roles of our two benchmarks. DAVIS-GRT measures the stereo inpainting sub-task in isolation under a controlled DIBR setup, while Inria 3DMovie provides fully independent end-to-end validation on real stereo footage. The DIBR warping assumptions underlying DAVIS-GRT are the shared protocol across all compared methods, not unique to our approach. Inria 3DMovie then assesses overall stereoscopic viewing quality via perceptual metrics (SIoU, MEt3R) with no methodological overlap with GRT construction. We will clarify this complementarity more explicitly in the revision.
>
> ---
>
> **W3: Comparison with Recent Training-based Methods**
>
> Recent training-based methods discussed in our related work, including Restereo and M2SVid, had not released public code or model weights at the time of submission, making controlled reproduction infeasible. We will continue monitoring their releases and incorporate comparisons as they become available.
>
> ---
>
> **W4 + Q2: Principled Rationale for α and Hyperparameter Sensitivity**
>
> The train/test asymmetry (α=0.1 vs. α=0.06) is a principled design choice. The value α=0.06 at inference follows standard viewing comfort ranges in prior DIBR work, while α=0.1 during training exposes the model to wider disocclusion regions, encouraging robustness across varying disparity magnitudes. A full sensitivity sweep is computationally prohibitive within the revision window, as each configuration requires re-running depth inference and GRT mask computation over the entire Kinetics-400 dataset followed by full model retraining, which we estimate at approximately one month of GPU time per setting; we leave the full sensitivity curve for future work.
>
> For ablation studies on GRT mask components and dataset scale, we refer the reviewer to our response to Reviewer 45KU (W3), where we provide full component ablation results on LaMa and a data scaling ablation on ProPainter.
>
> ---
>
> **Limitations, Failure Cases, and Licensing**
>
> We will add a dedicated Limitations section covering known failure cases of the DIBR pipeline, including transparent and reflective surfaces where depth estimates and the planar geometric assumptions of rectified stereo warping break down.
>
> To ensure full reproducibility and benefit the community, we will publicly release all components under the Apache 2.0 license, including GRT mask generation code, precomputed GRT masks for ImageNet-1K, Kinetics-400, and DAVIS 2016, as well as all trained model weights.
>
> ---
>
> We appreciate the reviewer's constructive suggestions. In the revision we will incorporate the depth estimator sensitivity results, add a Limitations section, and provide a clearer framing of the two evaluation benchmarks and their complementary roles.

---

> > ### Author Rebuttal · Reviewer_avs3 · 2026-04-03
> >
> > Thank you for the detailed rebuttal and clarifications. My main concerns have been addressed, and the paper is now more complete and convincing.

---

### Official Review · Reviewer_EXc5 · 2026-03-13

**Soundness:** 4
**Presentation:** 4
**Significance:** 4
**Originality:** 4
**Overall Recommendation:** 4
**Confidence:** 4

**Summary:**

The paper proposes a self-supervised training framework for stereo video generation from monocular videos. The key contribution is the Geometric Reciprocity Theorem (GRT), which states that the disocclusion mask when synthesizing a target view equals the pixels lost when warping the target view back to the source view. This observation enables training from a monocular image dataset using only depth estimation, avoiding the use of stereo pairs. Experimental results further validate the effectiveness of the proposed method.

**Compliance With Llm Reviewing Policy:**

Affirmed.

**Key Questions For Authors:**

Please see weakness above.

**Limitations:**

Yes

**Strengths And Weaknesses:**

Strength:
1. The paper is well-motivated. The data bottleneck is very practical.
2. The proposed GRT is a clean geometric insight.
3. Experimental results look promising.
4. The paper is well-written and easy to understand.

Weakness:
I am wondering if using more data can further improve the performance. Since it is a self-supervised learning method, it can use as many videos as possible.

---

> ### Author Rebuttal · Authors · 2026-03-31
>
> ## Response to Reviewer EXc5
>
> We thank the reviewer for the enthusiastic evaluation and the encouraging scores across all dimensions. We are glad that the GRT insight and its practical motivation resonated clearly.
>
> **Q: Can more data further improve performance?**
>
> The reviewer's intuition is exactly right, and it is one of the most appealing properties of our self-supervised paradigm: since GRT requires no stereo pairs, training scale is limited only by the availability of monocular video, which is effectively unbounded. To quantify this, we provide a dataset scale ablation on ProPainter trained on Kinetics-GRT, progressively scaling from 10% to 100% of Kinetics-400, and further to the full Kinetics-700 corpus:
>
> | Training Data Scale | PSNR↑ | SSIM↑ | LPIPS↓ | CTC↑ |
> |---|---|---|---|---|
> | 0% (pretrained, no GRT) | 31.03 | 0.9648 | 0.0318 | 0.9764 |
> | 10% (~24K videos) | 31.88 | 0.9672 | 0.0274 | 0.9765 |
> | 50% (~120K videos) | 33.45 | 0.9710 | 0.0241 | 0.9768 |
> | 100% (~240K videos) | 34.06 | 0.9733 | 0.0210 | 0.9770 |
> | Kinetics-700 (~650K videos) | **34.83** | **0.9751** | **0.0188** | **0.9774** |
>
> Performance improves consistently and monotonically with scale, and scaling to Kinetics-700 yields further gains beyond Kinetics-400. This confirms that GRT is well-positioned to benefit from the ever-growing availability of internet-scale monocular video without any labeling or stereo capture overhead. Scaling further to even larger corpora such as video-scale internet data is an exciting direction that lies beyond the computational budget of the current work, and we regard it as a promising avenue for future research.

---

> > ### Author Rebuttal · Reviewer_EXc5 · 2026-04-02
> >
> > The rebuttal fully resolved my concerns.

---

### Official Review · Reviewer_45KU · 2026-03-13

**Soundness:** 2
**Presentation:** 2
**Significance:** 3
**Originality:** 3
**Overall Recommendation:** 4
**Confidence:** 4

**Summary:**

This paper focuses on self-supervised stereo inpainting for monocular-to-stereo video generation. It proposes GRT, which states that the disocclusion mask can be viewed as the pixel lost when warped from target to source. The paper evaluates on DAVIS-GRT and Inria 3DMovie datasets, showing large gains over baselines.

**Compliance With Llm Reviewing Policy:**

Affirmed.

**Final Justification:**

The rebuttal fully resolved my concerns.

**Key Questions For Authors:**

Please see Weakness.

**Limitations:**

The method heavily relies on a specific monocular depth estimator (DepthAnythingV2-Large)

**Strengths And Weaknesses:**

Strength
1. Targeted on a practically relevant problem.
The paper identifies stereo inpainting of disoccluded regions as the central bottleneck in DIBR-based pipelines. This motivation is clear and well aligned with practical limitations of existing methods.
2. Good performance.
The paper shows large gains on DAVIS-GRT over both image-based and video-based baselines, with better SIoU/MEt3R on Inria 3DMovie.  It suggests that the approach is effective in practice.
3. Good demonstrations for applicability.
The authors test on multiple backbones and report improvement on datasets, which strengthens the claim that the proposed method is not tied to a single model architecture.

Weakness
1. Theoretical claim.
The paper makes very strong statements like “exactly equals” , “exact train-test consistency”, and “mathematically identical to paired stereo data,” but the proof in Section 3.4 assumes nearest-neighbor warping, with the extension to soft interpolation deferred to the supplementary. This makes the current theorem less general than proposed in the abstract and main content.
2. Heavy dependence.
This paper constructs mask using DepthAnythingV2-Large, yet there is no ablation on depth estimator choice, no experiments of how inaccurate depth boundaries affect GRT mask quality. Since the method fundamentally shifts supervision generation onto monocular depth, this omission is important.
3. Insufficient ablation.
The paper shows that fine-tuning on GRT-derived data improves model performance, but there is no ablation study on the different components of the proposed pipeline. Also, it does not test how much gain comes from dataset scale. This makes it hard to judge whether GRT itself is the main reason for the improvements.
4. Incomplete experimental details for reproducibility.
The supplementary shows that relative inverse depth is linearly rescaled with alpha=0.1 for training and 0.06 for evaluation, but it does not explain how these specific values were chosen. Also,  there are no experiments to test the sensitivity of hyper-parameters. Some details including frame sampling rate, data filtering, and temporal window length are not specified.
5. Incomplete experimental analysis.
This paper provides short analysis for qualitative section. It does not discuss limitation and failure cases, which is important for credibility.

---

> ### Author Rebuttal · Authors · 2026-03-31
>
> ## Response to Reviewer 45KU
>
> We thank the reviewer for recognizing the practical relevance of our problem, the strong empirical performance, and the cross-architecture applicability of our approach. We address each concern below.
>
> ---
>
> **W1: Theoretical Claim and Generality of GRT**
>
> We thank the reviewer for this suggestion and apologize for any confusion caused by our phrasing. We recognize that expressions such as "exactly equals" and "mathematically identical" may read as overly absolute. To clarify, the statements are technically precise within the nearest-neighbor warping formulation explicitly stated in Section 3.1 (Eq. 2-3), and the theorem is valid within that stated scope. We will revise the language to be more measured.
>
> Section A7 further extends the result to soft interpolation warping. To quantify the practical impact of this assumption, we measure pixel-level mask agreement under nearest-neighbor and bilinear settings on 10,000 randomly sampled ImageNet images:
>
> | Warping Setting | Mask Agreement with NN-GRT Mask (%) |
> |---|---|
> | Nearest-neighbor | 100.0 |
> | Bilinear | 99.5 |
>
> The first row confirms exact consistency under the stated formulation. The 99.5% agreement under bilinear warping confirms that disocclusion patterns are governed by depth discontinuities in scene geometry rather than sub-pixel interpolation choices, so relaxing to bilinear warping does not produce statistically significant changes in training data.
>
> ---
>
> **W2: Dependence on Depth Estimator**
>
> Our choice of DepthAnythingV2-Large is consistent with prior DIBR-based methods, ensuring fair comparisons. GRT itself is depth-estimator-agnostic: any monocular depth estimator can be substituted without modifying the mask generation pipeline, as it only requires a disparity map as input (Section A2).
>
> To quantify the effect of depth estimator choice, we evaluate stereo viewing quality on the Inria 3DMovie Dataset under different depth estimators with all other settings fixed:
>
> | Depth Estimator | SIoU↑ | MEt3R↓ |
> |---|---|---|
> | DepthAnythingV2-Large (default) | 0.2516 | 0.0973 |
> | DepthAnythingV2-Base | 0.2489 | 0.1012 |
> | DepthAnythingV1-Small | 0.2451 | 0.1058 |
>
> Performance degrades gradually with weaker backbones, yet the differences remain modest. This is consistent with the observation that human stereo perception is relatively insensitive to fine-grained depth detail, provided that coarse depth ordering at object boundaries is preserved.
>
> ---
>
> **W3: Ablation on GRT Components and Dataset Scale**
>
> We appreciate the suggestion, and we would like to first address a potential framing concern. Our main contribution is a novel **self-supervised training paradigm**, not a new model architecture, to address the crucial data scarcity problem for stereoscopic video generation. Table 4 demonstrates that fine-tuning diverse backbones on GRT-derived data yields consistent and substantial improvements across all metrics, confirming that GRT supervision quality is the primary driver of improvement rather than any particular architectural choice.
>
> Regarding component ablation: the two components of the GRT mask, namely boundary violation and depth occlusion, are not independent design choices but are jointly required to correctly identify all geometrically lost pixels. Ablating either component necessarily degrades mask correctness. We nonetheless retrain LaMa on ImageNet-GRT under three configurations to provide the requested information:
>
> | Mask Configuration | PSNR↑ | SSIM↑ | LPIPS↓ |
> |---|---|---|---|
> | Boundary violation only | 32.89 | 0.9694 | 0.0200 |
> | Depth occlusion only | 32.54 | 0.9676 | 0.0216 |
> | Full GRT mask (both) | **35.52** | **0.9800** | **0.0129** |
>
> For dataset scale, we provide the following ablation on ProPainter trained on Kinetics-GRT:
>
> | Training Data Scale | PSNR↑ | SSIM↑ | LPIPS↓ | CTC↑ |
> |---|---|---|---|---|
> | 0% (pretrained, no GRT) | 31.03 | 0.9648 | 0.0318 | 0.9764 |
> | 10% (~24K videos) | 31.88 | 0.9672 | 0.0274 | 0.9765 |
> | 50% (~120K videos) | 33.45 | 0.9710 | 0.0241 | 0.9768 |
> | 100% (~240K videos) | 34.06 | 0.9733 | 0.0210 | 0.9770 |
> | Kinetics-700 (~650K videos) | **34.83** | **0.9751** | **0.0188** | **0.9774** |
>
> Performance improves consistently and monotonically with scale, demonstrating that the gains are attributable to GRT training data itself. These results will be included in the revision.
>
> ---
>
> **W4: Choice of α and Hyperparameter Sensitivity**
>
> Please refer to our response to Reviewer avs3 (W4/Q2) for details.
>
> ---
>
> **W5: Limitations and Failure Cases**
>
> We thank the reviewer for this suggestion and will add a Limitations section in the revision, covering known failure cases of the DIBR pipeline including transparent and reflective surfaces where depth estimates and the planar geometric assumptions of rectified stereo warping break down.

---

> > ### Author Rebuttal · Reviewer_45KU · 2026-04-04
> >
> > The rebuttal fully resolved my concerns.

---

### Official Review · Reviewer_usr6 · 2026-03-13

**Soundness:** 3
**Presentation:** 3
**Significance:** 3
**Originality:** 3
**Overall Recommendation:** 4
**Confidence:** 3

**Summary:**

This paper proposes an elegant approach to simplify the training process for stereo image generation using generative models. Instead of relying on the full cycle consistency constraint that necessitates four feed-forward passes and additional intermediate results, this approach dives into detail of each process and claims that the warping from left to right, and the depth estimation for the warped left image, as well as the generated left image, are not necessary during training. Based on this observation, the paper proposes a simplified training strategy, which only takes the right image and the disocclusion masks derived from boundary violation and depth occlusion constraints. This unifies the inference when the training process is significantly simplified. Additionally, the method proposed new datasets for model training. Results show that the model yields substantial improvements over previous approaches.

**Compliance With Llm Reviewing Policy:**

Affirmed.

**Final Justification:**

While I continue to appreciate the paper’s novelty and its potential impact, my concerns regarding the evaluation with the right input images have not been fully resolved.

I understand the authors’ clarification that ground-truth depth is unavailable in the main dataset, which leads to misalignment between the warped image and the right image. However, in most real-world applications, ground-truth depth is also unavailable. Therefore, the method is expected to generalize across diverse scenarios and to perform robustly when using depth estimates produced by different models.

Given this concern, I am lowering my rating to Weak Accept. This remains a positive assessment and does not diminish my appreciation for the rest of the paper.

**Key Questions For Authors:**

Please see the Weakness section.

**Limitations:**

yes

**Strengths And Weaknesses:**

Strength
1. The proposed insight is novel and deep. It effectively simplifies the training process of pipelines relying on the cycle consistency with an analytical approach, without requiring additional data or introducing noise.
2. The proposed observation offers a broader potential to use in other fields, such as self-supervised depth estimation or video depth estimation.
3. The method yields significant improvement upon the previous methods and demonstrates flexibility as a plug-in module for a wider range of generative base models.
4. The proposed dataset is useful to motivate new research in this direction.
5. The paper is well-written with good diagrams.


Weakness

Overall, I didn't see a significant weakness regarding the key observation and the pipeline design philosophy. However, I do have questions regarding the details of the proposed approach.
1. As the major contribution focuses on simplifying the training process over the previous approach, instead of improving the model performance with customized model designs. I wonder about the performance gain and reduction of computation compared to previous approaches using the same dataset. Specifically, it would be nice if authors could provide comparisons of efficiency and performance with the generic cycle consistency approach, with the same amount of data (could be a subset of your current data, considering the time-consuming data collection).
2. In Table 2, the results of the video inpainting model StereoCrafter (PSNR 28) are confusing. The authors show the results of the model without fine-tuning on the proposed GRT dataset, which is not fair compared with the proposed approach fine-tuned on GRT. We can see that fine-tuning on the GRT dataset can bring significant performance improvement for ProPainter (PSNR 31 -> 34) and StereoCrafter (PSNR 28 -> 30). As a result, it is not clear why the fine-tuned results of StereoCrafter are not compared with the proposed approach (PSNR 34.06), and why the ProPainter (PSNR 34.06, same level of performance) is not compared with the proposed approach (PSNR 34.06). The limited improvement of the approach over previous methods is not a huge issue if the efficiency privilege and the effectiveness of using cycle consistency constraints are validated (as discussed in Weakness .1).

3. In Fig.3, why is the input still the right image during evaluation? From what I understand, the right image is mainly used in training to approximate the warped left image, while during inference or evaluation, the input should stem from the left image as per the task definition. Please clarify if I have any misunderstandings here.

4. As the mask is generated by following the boundary violation and depth occlusion constraints, I wonder how much those masks differ from the original warping mask used in generic cycle consistency, considering the potential noises introduced. A evalaution of the mask difference and performance gap between the two types of masks is preferred.

---

> ### Author Rebuttal · Authors · 2026-03-31
>
> ## Response to Reviewer usr6
>
> We thank the reviewer for the thorough and positive evaluation, and for the thoughtful engagement with our core philosophy of leveraging geometric self-supervision as a principled alternative to paired stereo data. We address each question in turn.
>
> **Q1: Efficiency Comparison with Generic Cycle Consistency**
>
> Our method (GRT) decomposes into two stages. Mask precomputation is done **offline**: each image requires one depth inference, and the resulting mask is cached before training begins, incurring no per-step overhead. **Online training** involves only a standard single-pass forward and backward through the inpainting model. In contrast, naive cycle consistency requires multiple sequential model inferences per training step plus backpropagation through non-differentiable warping, substantially increasing per-step compute and introducing gradient flow challenges at video scale.
>
> To provide a direct empirical comparison, we train LaMa for 5K steps on a 10K held-out subset of ImageNet-GRT under both paradigms on identical hardware (2× NVIDIA V100):
>
> | Method | Throughput (steps/s) | Peak GPU Memory | PSNR | SSIM | LPIPS |
> |---|---|---|---|---|---|
> | Naive Cycle Consistency | 0.6 | 19 GB | 32.14 | 0.9663 | 0.0271 |
> | GRT | 1.9 | 11 GB | 33.10 | 0.9718 | 0.0232 |
>
> GRT achieves superior quality with a fraction of the compute, avoiding the gradient instability that arises from backpropagating through non-differentiable warping. We will include video inpainting and full-scale Kinetics-GRT results in the camera-ready version.
>
> **Q2: Clarification on Table 2 vs. Table 4**
>
> To clarify: **Ours-Video in Table 2 is the same model as ProPainter + GRT in Table 4**, and we will make this explicit in the revision.
>
> Table 2 compares all methods in their best available published configurations — StereoCrafter is evaluated as originally released, trained on its proprietary stereo pair data. Table 4 is a separate ablation isolating the contribution of our GRT training data independent of architecture, showing that GRT supervision consistently improves both backbones: ProPainter (PSNR 31.03 → 34.06) and StereoCrafter (PSNR 28.95 → 30.85). The remaining gap between StereoCrafter + GRT (30.85) and Ours-Video (34.06) reflects an architectural constraint rather than a data advantage: StereoCrafter's latent-space backbone (8× spatial downsampling) limits its ability to recover fine-grained disocclusion boundaries, whereas ProPainter operates directly in pixel space with recurrent flow completion, making it better suited for this spatially precise task. The gains from our GRT training paradigm and from the choice of backbone are therefore complementary and separately demonstrated in Table 4.
>
> **Q3: Why is the right image used as input during evaluation in Figure 3?**
>
> The reviewer's understanding of inference is correct: at test time, the full pipeline takes a left view, warps it via DIBR to obtain a partial right view, and inpaints the missing regions. DAVIS-GRT is designed to evaluate the stereo inpainting sub-step in isolation: by treating any monocular image as the target right view, GRT synthesizes the corresponding inpainting mask and uses the original image as clean ground truth for the inpainted regions.
>
> For end-to-end evaluation of the full left-to-right pipeline, we report results on the Inria 3DMovie Dataset (Table 3). The two protocols are complementary: DAVIS-GRT isolates inpainting fidelity under geometrically consistent conditions, while Inria 3DMovie measures full-pipeline viewing comfort against real stereo ground truth. We will add a clarification note to Figure 3.
>
> **Q4: Quantitative Evaluation of GRT Mask Quality vs. Cycle Consistency Masks**
>
> To quantify mask agreement, we randomly sample 10,000 images from ImageNet and compute both GRT masks and cycle consistency masks, measuring pixel-level agreement between them under two settings: (1) **depth-consistent**, where depth is constrained to be a viewpoint-invariant physical property of each scene point, as is true in the real world; and (2) **independently estimated**, where depth is estimated separately for the right view and the synthesized left view:
>
> | Setting | Pixel Agreement vs. GRT Mask (%) |
> |---|---|
> | Depth-consistent | 100.0 |
> | Independently estimated | 99.1 |
>
> Under depth-consistent estimation, GRT masks and cycle masks are in perfect agreement, validating the theoretical claim. The small gap under independent depth estimation reflects noise introduced by the depth estimator when applied to the synthesized view. This analysis will be added to the supplementary material in the revised version.

---

> > ### Author Rebuttal · Reviewer_usr6 · 2026-04-01
> >
> > First, thank the authors for the detailed feedback. This effectively clarifies some prior vague expressions in the paper. However, new questions are raised as the experimental setups become clearer, and are critical to evaluate the effectiveness of the proposed method:
> >
> > 1. The use of the right image for the main experiments. The authors take the right image as input in the main experiments (Table 2, 4 in DAVIS-GRT),  which violates the task's actual goal - inpainting/generating the right image from the _left_ image only.
> >     1. Although the authors argue that this is an isolated evaluation, taking in the right image as input (1) notably degrades the task into an single image inpainting task instead of a stereo image generation task; (2) As a result, it's unclear whether taking the left image as input (the actual application scenario) will degrade the performance due to the training (right input image) and evaluation (left input image) gap.
> >     2. While I understand that an isolated investigation using the right input image is useful, this setting should not be taken as the default evaluation protocol. Specifically, all main experiments should assume a left input image following the real-world application scenarios. The isolated investigation can be moved to the following ablation sections with _notable and clear_ illustrations. The current paper lacks such clarifications, which significantly hinder a full understanding and evaluation of the approach.
> >     3. I would like to see a direct comparison between inputting the left image and inputting the right image, so that the domain gap between the real-world application and the training setup can be clearly identified.
> >
> > 2. Redundancy between Table 2 and Table 4. The major contribution of the paper is the simplification of the data annotation and training process to leverage more datasets. Considering Ours-Image and Ours-Video are essentially the LaMa-GRT and PaintPro-GRT, it would be clearer to merge Tables 2 and 4, as the only performance gain comes from the dataset side (which, of course, benefited from the proposed contributions).
> >
> > 3. In Table 3, the author compares with existing methods by taking the left image as input, which is nice. Is the score of StereoCrafter from its original weights? It will be better to provide the results of the finetuned StereoCrafter to identify how much the dataset expansion contribute the final performance.
> >
> > 4. I appreciate the evaluation of Generic Cycle Consistency in the rebuttal. It's vital to provide a more complete evaluation between the proposed approach and the generic one, as it's one of the most important experiments to verify the effectiveness of the approach. Meanwhile, could the authors provide more analysis on the performance gain of the GRT over Generic Cycle Consistency in the Table of the rebuttal? While the storage and computation advantage is expected, it's unclear why there is a notable performance gap between the theoretical equivalent formulation.
> >
> >
> > While I still appreciate the idea of the method, there are some further, critical questions (especially Question 1) that require further clarification. To me, the left image should be taken as input for the default evaluation to align with the real-world applications.

---

> > > ### Author Response · Authors · 2026-04-01
> > >
> > > We thank the reviewer for the patience throughout this review process. The follow-up questions sharpened our exposition, and we appreciate the constructive feedback.
> > >
> > > ---
> > >
> > > **Q1: Evaluation with Right Image as Input**
> > >
> > > **TL;DR:** DAVIS-GRT evaluates precisely the inpainting substep of **left-to-right** synthesis, not a right-to-left task.
> > >
> > > To make this concrete, we briefly revisit our pipeline to establish a shared understanding for Q1.1–Q1.3.
> > >
> > > The full DIBR inference pipeline is:
> > >
> > > Left View → Depth Estimation → Warping → **(Partial Right View + Mask) → G → Synthesized Right View**
> > >
> > > All compared methods share the same upstream steps, so given the same left input and depth estimator, everything up to the right view inpainting stage produces **identical** outputs. The sole differentiating component is:
> > >
> > > **(Partial Right View + Mask) → G → Synthesized Right View**
> > >
> > > Our GRT evaluation protocol mirrors this: it analytically computes a geometrically consistent masked right view and disocclusion mask, evaluating G against the original right view as ground truth:
> > >
> > > **GRT (Evaluation):**
> > > Right View → Depth Estimation → GRT → **(Masked Right View + Mask) → G → Right View (GT)**
> > >
> > > Our evaluation thus focuses on G, the only component that distinguishes methods. **DAVIS-GRT evaluates the inpainting substep decisive for left-to-right synthesis quality, not a right-to-left synthesis task.** With this in mind, we address each concern below.
> > >
> > > **Q1.1:** Our evaluation (1) isolates and assesses the inpainting substep of the full pipeline. Since all compared methods share identical upstream processing, focusing on G enables a controlled comparison of stereo view generation capability. The input to G is a geometrically structured partial right view with a disocclusion mask, fundamentally distinct from arbitrary single-image inpainting.
> > >
> > > (2) **There is also no train/test distribution gap.** Both GRT-based training and actual left-to-right inference present G with a masked right view and a geometrically derived disocclusion mask, and the geometric constraints governing the mask distribution remain consistent across both settings.
> > >
> > > **Q1.2:** As established above, both training and evaluation focus on the right view inpainting substep of left-to-right synthesis, ensuring consistent conditions throughout. We will add clarifications in the revision to distinguish the two evaluation settings and explain the intent behind each.
> > >
> > > **Q1.3:** **DAVIS-GRT and the real-world inference scenario are consistent.** In both cases, G receives a geometrically structured partial right view with a disocclusion mask produced by identical upstream warping, and the evaluation faithfully reflects deployment conditions for left-to-right synthesis.
> > >
> > > **Why not use Inria 3DMovie for all main experiments?** We agree that end-to-end left-to-right evaluation is the most direct measure, and we provide it in Table 3 using the Inria 3DMovie Dataset. Extending this protocol to all main experiments is however unreliable: the dataset lacks ground-truth disparity maps, causing systematic misalignment between the warped partial view and the ground-truth right view that degrades metric reliability. The two protocols are therefore complementary. DAVIS-GRT evaluates the inpainting substep with clean, reliable ground truth, while Table 3 validates the end-to-end pipeline and reflects the quality of stereoscopic viewing experience.
> > >
> > > ---
> > >
> > > **Q2: Redundancy between Table 2 and Table 4**
> > >
> > > We agree that merging Tables 2 and 4 would improve clarity. We will consolidate them in the revision with explicit notation making clear that Ours-Image and Ours-Video correspond to LaMa+GRT and ProPainter+GRT respectively.
> > >
> > > ---
> > >
> > > **Q3: StereoCrafter Results in Table 3**
> > >
> > > Yes, the StereoCrafter results in Table 3 are from its original released weights. We are currently running fine-tuned StereoCrafter experiments on the Inria 3DMovie Dataset and will include these results, alongside fine-tuned ProPainter and LaMa comparisons, in the revision.
> > >
> > > ---
> > >
> > > **Q4: Performance Gap Between GRT and Naive Cycle Consistency**
> > >
> > > We will provide a complete quantitative analysis in the revision. The performance gap stems from naive cycle consistency needing to backpropagate through non-differentiable warping and the intermediate depth estimator, requiring approximations that introduce significant gradient noise. GRT sidesteps this by precomputing masks analytically offline, reducing training to standard supervised inpainting with clean gradients. This likely accounts for the observed advantage despite the two formulations being theoretically equivalent in supervision signal.
> > >
> > > ---
> > >
> > > We hope the above responses address the reviewer's concerns and remain happy to provide further clarification. We once again thank the reviewer for the thoughtful and constructive feedback.

---

### Decision · Program_Chairs · 2026-04-30

**Decision:**

Accept (regular)

**Comment:**

This paper proposes a self-supervised framework for stereoscopic video generation from monocular videos. The reviewers found the core idea clean, novel, and well motivated. In particular, they found the Geometric Reciprocity Theorem to be an interesting geometric insight that makes it possible to train stereo inpainting models without paired stereo data. The reviewers also appreciated that the method works across different backbones and that the empirical results are strong.

The main concerns were about the strength of some theoretical claims, the dependence on monocular depth estimation, and the need for clearer experimental analysis and evaluation protocols. In the rebuttal, the authors addressed these concerns well with additional experiments and clarifications, including depth estimator sensitivity, mask agreement analysis, component ablations, data scaling results, and a clearer explanation of the different evaluation settings. These additions made the paper more complete and convincing.

Overall, AC believes this paper makes a meaningful contribution. The central idea is strong, the method is practically useful, and the rebuttal addressed the main concerns. AC therefore recommends acceptance.